# Diterpenes Specially Produced by Fungi: Structures, Biological Activities, and Biosynthesis (2010–2020)

**DOI:** 10.3390/jof8030244

**Published:** 2022-02-28

**Authors:** Fa-Lei Zhang, Tao Feng

**Affiliations:** School of Pharmaceutical Sciences, South-Central University for Nationalities, Wuhan 430074, China; flzhang@mail.scuec.edu.cn

**Keywords:** diterpenes, isolation, structure, biological activity, biosynthesis

## Abstract

Fungi have traditionally been a very rewarding source of biologically active natural products, while diterpenoids from fungi, such as the cyathane-type diterpenoids from *Cyathus* and *Hericium* sp., the fusicoccane-type diterpenoids from *Fusicoccum* and *Alternaria* sp., the guanacastane-type diterpenoids from *Coprinus* and *Cercospora* sp., and the harziene-type diterpenoids from *Trichoderma* sp., often represent unique carbon skeletons as well as diverse biological functions. The abundances of novel skeletons, biological activities, and biosynthetic pathways present new opportunities for drug discovery, genome mining, and enzymology. In addition, diterpenoids peculiar to fungi also reveal the possibility of differing biological evolution, although they have similar biosynthetic pathways. In this review, we provide an overview about the structures, biological activities, evolution, organic synthesis, and biosynthesis of diterpenoids that have been specially produced by fungi from 2010 to 2020. We hope this review provides timely illumination and beneficial guidance for future research works of scholars who are interested in this area.

## 1. Introduction

Fungi are widely distributed in terrestrial environments, freshwater, and marine habitats; more than one million distinctive fungal species exist, but only approximately 100,000 of these have been classified [1]. These eukaryotic microbes produce specialized metabolites that participate in a variety of ecological functions, such as quorum sensing, chemical defense, allelopathy, and maintenance of symbiotic interactions [2]. There are more than 40,000 terpenoid compounds in nature, which compose the largest family of natural products [3]. Terpenoids exist in all domains of life, but are particularly prevalent in plants, fungi, and marine invertebrates, and are essential constituents of secondary metabolism [3,4].

Diterpenoids are a class of C20 compounds derived from isoprenoid precursor geranylgeranyl diphosphate (GGPP) under the catalysis of diterpene synthases (DTSs) [5,6,7,8,9,10,11]. Prenyltransferase (PT) and terpene synthase (TPS) are key enzymes in the formation of the basic carbon skeletons of terpenoids [8,12]. The PT enzymes determine the prenyl carbon chain length, whereas the TPS enzymes generate the structural complexity of the molecular scaffolds, forming various ring structures [8]. Fungi are among the most important microbial resources for drug discovery, owing to their capability to produce structurally diverse and biologically important secondary metabolites [13,14]. It is also well known that fungi possess extraordinary biosynthetic gene clusters that may encode highly diverse biosynthetic pathways of natural products [15,16,17,18]. 

Between 2010 and 2020, about 400 fungal-specific diterpenes have been reported. In addition to 172 cyathane diterpenes reviewed by Bailly et al. [19] and Gao et al. [20], a total of 232 diterpenes were collected in this review ([Fig jof-08-00244-ch001]). These diterpenoids are mainly tricyclic or tetracyclic skeletal structures such as cyathane-type, fusicoccane-type, guanacastane-type, and harziene-type diterpenoids ([Fig jof-08-00244-ch001]). Judging from the distribution of fungal diterpenoid resources, the diterpenes from the genera *Trichoderma*, *Penicillium*, *Cyathus*, *Hericium*, and *Crinipellis* account for 60% of the total ([Fig jof-08-00244-ch002]). In addition, systematic studies on the chemical constituents of fungi have shown that a large number of fungal diterpenoids exhibited significant biological functions such as anti-inflammatory, cytotoxic, antimicrobial, and antiviral activities ([Fig jof-08-00244-ch003]). For instance, the semi-synthetic pleuromutilin analogues tiamulin **193** and valnemulin **194** have been used for over three decades as antibiotics to treat economically important infections in swine and poultry [21,22,23,24,25].

Consequently, a wealth of novel skeletons, biosynthetic pathways, and bioactivities have provided new opportunities for drug discovery, genome mining, enzymology, and chemical synthesis. During the period covered in this review, there have been several more specialized reviews of fungal metabolites [26,27,28], including benzene carbaldehydes [29], trichothecenes [30,31], protoilludane sesquiterpenoids [32], meroterpenoids [33,34,35], meroterpenoid cyclases [36], terpenoids [37], and natural product biosynthetic genes and enzymes of fungi [17,18,38,39]. In addition, the isolation and chemistry of diterpenoids from terrestrial sources have been summarized [40]. In this review, we provide an overview of diterpenoids that were specially produced by fungi during the period from 2010 to 2020, and focus on their structures, biological activities, and biosynthesis, and we also conduct an evolutionary analysis.

In particular, literature investigation of known databases such as PubMed and Web of Science was conducted from 2010 to July 2020 using the keywords “diterpenes/diterpenoids” paired with “fungi”, “fungal diterpenoids” paired with “structure elucidation”, or “fungal diterpenoids” paired with “biosynthesis”. There were no language restrictions imposed. The references were further scrutinized and, finally, 210 references were selected. The data inclusion criteria included: (1) diterpenes/diterpenoids isolated from fungi, (2) carbon skeleton obtained only from fungi or rarely from other sources, (3) studies on the biological activities of diterpenes/diterpenoids and their derivatives that had been carried out in vitro or in vivo, (4) studies on the biosynthesis of diterpenes/diterpenoids and their derivatives. The data exclusion criteria included: (1) carbon skeleton of diterpenes/diterpenoids obtained in abundance from other sources, such as plants, bacteria and so on, (2) duplication of data and titles and/or abstracts not meeting the inclusion criteria.

## 2. Cyathane



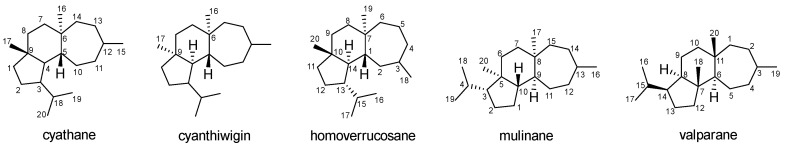



Cyathane diterpenes are a group of natural products that possess unusual, angularly fused 5/6/7 tricyclic cores, and they are characteristic of certain basidiomycete species including *Cyathus*, *Hericium*, and *Sarcodon* (Figure 1). For example, there have been more than 170 compounds isolated from fungi such as *Cyathus africanus* and *Hericium erinaceus* [19,20,41]. These compounds have a common biosynthetic precursor and can be produced via biosynthesis, hemi-synthesis, or total synthesis [42,43,44,45,46,47]. The cyathane diterpenoids include the classes of cyathins, striatins, sarcodonins, scabronines, and erinancines, according to their origins. Among them, the striatals, striatins, and erinacines, called cyathane-xylosides, which represent an unusual group of cyathane diterpenoids attached to a modified pentose (*D*-xylose) moiety, have been isolated from cultures of *Cyathus* and *Hericium* species [20]. The cyathane structure is different from the homoverrucosane, mulinane, and valparane diterpenoids which also possess a 5/6/7 tricarbocyclic system [48,49]. The cyathanes are most similar to cyanthiwigins and can be differentiated by the orientation of the angular methyl groups, mainly present in some sponges [50,51,52,53,54,55,56,57,58,59]. These compounds display a diverse range of biological activities, including anticancer, antimicrobial, anti-MRSA (methicillin-resistant *Staphylococcus aureus*), anti-inflammatory, anti-proliferative, and nerve growth factor (NGF)-like properties [19,20,60,61]. An overview of cyathane-type diterpenes including isolation, structure diversity, synthesis, and bioactivity has been reviewed by Bailly et al. [19] and Gao et al. [20]. Therefore, in this review, we no longer summarize the details of cyathane diterpenoids.

To understand the source genera of cyathane diterpenoids, we performed a phylogenetic analysis by using the maximum likelihood method and the general time reversible model [62,63,64] for all the species involved in the reviews by Bailly et al. [19] and Gao et al. [20]. The results show that source genera are grouped based on their regiospecificity, i.e., genera *Cyathus*, *Hericium*, and *Sarcodon* were clustered into different clades (Figure 1). Taxonomically, *Cyathus africanus*, *C. hookeri*, *C. gansuensis*, *C. subglobisporus*, *C. stercoreus*, and *C. striatus* all belonged to the genus *Cyathus*. They were close to each other, and first, they gathered into one branch, then, they gathered into one branch with *Strobilurus tenacellus* of the genus *Strobilurus*, and finally gathered into one branch with other genera (Figure 1). *C. earlei* and *C. helenae* also belonged to the genus *Cyathus*, they were close to each other, and first, they gathered into one branch, then, they gathered into one branch with *Gerronema albidum* of the genus *Gerronema*. Similarly, *Hericium erinaceus*, *H. flagellum*, and *Hericium* sp. WBSP8, *Sarcodon scabrosus*, *S. glaucopus*, and other species were close to each other. Existing studies have shown that most fungal metabolites are encoded by biosynthetic gene clusters (BGCs) [17]. The natural product BGCs of species in the same genus tend to be highly homologous, and BGC functional divergence gives rise to the evolution of new secondary metabolites, indicating that species-level sampling in these three genera for natural products mining will yield significant returns for cyathane diterpenoids discovery.

## 3. Cyclopiane



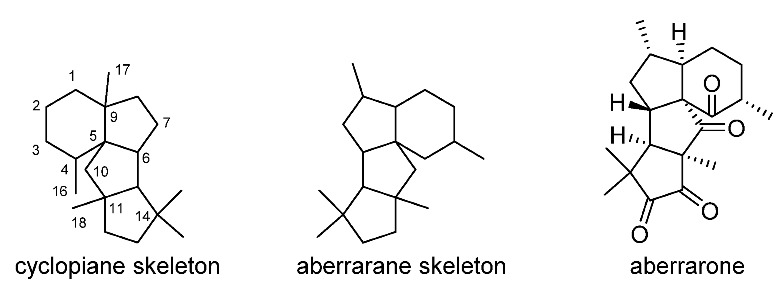



Cyclopiane diterpenoids comprise a class of tetracyclic diterpenes with unique scaffolds. They are characterized by a highly fused 6/5/5/5 ring system. The structural variations of cyclopiane diterpenoids are mainly owing to oxidation occurring at various sites to generate hydroxy groups [65]. In general, cyclopiane diterpenoids have mainly been isolated from different species of the genus *Penicillium* (Figure 2) and have been classified into two groups according to the functionality at C-1, i.e., conidiogenols and conidiogenones. The former featured with a hydroxy group at C-1, while the later possessed a carbonyl group at C-1 [66]. Specifically, *Penicillium commune* MCCC 3A00940, *P.* sp. F23-2, *P.* sp. YPGA11, *P. cyclopium*, *P. roqueforti* IFM 48062, *P.* sp. TJ403-2, *P. chrysogenum* QEN-24S, and *Leptosphaeria* sp. XL026 have been reported to produce conidiogenol-type diterpenoids, while *P. commune* MCCC 3A00940, *P. chrysogenum* MT-12, *P.* sp. YPGA11, and *P. cyclopium* have been reported to produce conidiogenone-type diterpenoids (Figure 2). Structurally, cyclopiane diterpenoids differ from the aberrarane-type diterpenoid aberrarone, which has shown in vitro antimalarial activity against a chloroquine-resistant strain of the protozoan parasite *Plasmodium falciparum* isolated from the Caribbean sea whip *Pseudopterogorgia elisabethae* [67]. The molecular structure of aberrarone was established by spectral analysis and subsequently confirmed by X-ray crystallographic analysis. Some cyclopiane compounds exhibited pronounced biological activities, such as conidiation induction, cytotoxic, anti-inflammatory, antimicrobial, and antiallergic effects.



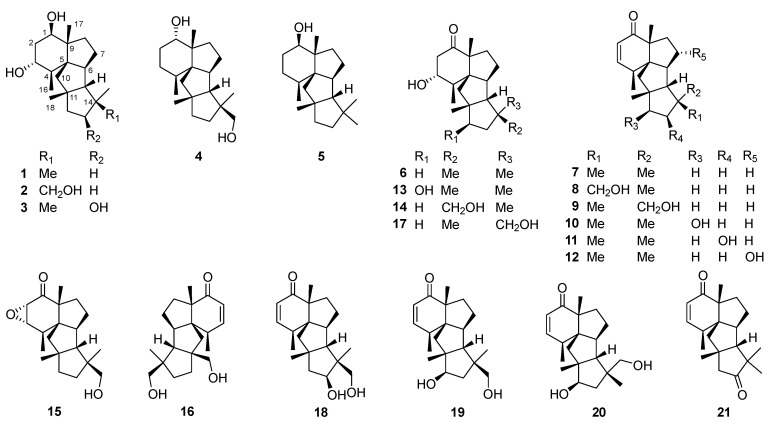



### 3.1. Conidiogenol Type

Conidiogenol **1** is a potent and selective inducer of conidiogenesis in the liquid culture of *Penicillium cyclopium* under non-nutrient limiting conditions [66]. Conidiogenol B **2** has been obtained from the deep-sea derived fungus *P. commune* MCCC 3A00940 [68]. Conidiogenols C **3** and D **4** have been isolated from a deep-sea derived fungus *P.* sp. YPGA11 [65].

The absolute structure of cyclopiane diterpenoids was first confirmed by Abe and co-workers, in 2018, with the aid of the crystal sponge method [69]. Using the genome-mining approach, a chimeric enzyme of prenyltransferase-diterpene synthase (PT-TS) discovered from *P. chrysogenum* MT-12 was designated as *P. chrysogenum* cyclopiane-type diterpene synthase (PcCS). The new diterpene alcohol metabolite **5** was produced after the gene heterologously expressed in *Aspergillus oryzae*, and the crystalline sponge method also revealed the absolute configuration of **5** [69]. The PT domain of PcCS first generated geranylgeranyl diphosphate (GGPP) from dimethylallyl pyrophosphate (DMAPP) and isopentenyl pyrophosphate (IPP) (Figure 1A). Then, GGPP was converted into **5** by a cyclization reaction catalyzed by the TS domain of PcCS (Figure 1B).

### 3.2. Conidiogenone Type

Conidiogenone **6**, first isolated from *Penicillium cyclopium*, was also an inducer of conidiation [66,70]. The biosynthetic pathway of (–)-conidiogenone **6** has been fully elucidated by the heterologous expression of biosynthetic genes in *Aspergillus oryzae* and by in vitro enzyme assay with ^13^C-labeled substrates [71]. After construction of deoxyconidiogenol by the action of bifunctional terpene synthases (PchDS gene obtained from *Penicillium chrysogenum*, and PrDS gene identified from *Penicillium roqueforti* showed significant homology to PchDS), one cytochrome P450 catalyzed two rounds of oxidation to furnish conidiogenone **6**. The cyclization mechanism catalyzed by terpene synthase, involving successive 1,2-alkyl shifts, was fully elucidated using ^13^C-labeled geranylgeranyl pyrophosphate (GGPP) as a substrate (Figure 2).

A series of new conidiogenone-type diterpenoids have been obtained from several *Penicillium* species including conidiogenones B–G **7**–**12** from the fungus *P.* sp. F23-2 [72], conidiogenones H **13** and I **14** from the endophytic fungus *P. chrysogenum* QEN-24S [73], conidiogenones J **16** and K **15** from the fungus *P. commune* [68], and conidiogenone L **17** from *P.* sp. YPGA11 [65]. Conidiogenone B **7** showed potent activity against methicillin-resistant *Staphylococcus aureus* (MRSA), *Pseudomonas fluorescens*, *P. aeruginosa*, and *Staphylococcus epidermidis* (each with a MIC value of 8 μg/mL) [73]. Conidiogenone C **8** showed potent cytotoxicity against HL-60 and BEL-7402 cell lines with IC_50_ values of 0.038 and 0.97 μM, and conidiogenone G **12** showed potent cytotoxicity against HL-60 cell line with an IC_50_ value of 1.1 μM [72]. Provoked by the novelty of structures and potent bioactivities, total syntheses of **1**, **6**, and **7** were achieved, which led to further determination of their absolute configurations [74]. 

Three new cyclopiane diterpenes 13*β*-hydroxy conidiogenone C **18** and 12*β*-hydroxy conidiogenones C **19** and D **20** have been isolated and identified from a sea sediment-derived fungus *Penicillium* sp. TJ403-2 [75]. Their absolute configurations were further established by X-ray crystallography experiment. Compounds **18**–**20** were evaluated for their anti-inflammatory activity against LPS-induced NO production, and compound **18** showed notable inhibitory potency with an IC_50_ value of 2.19 μM, which was three-fold lower than the positive control indomethacin (IC_50_ 8.76 μM). Further Western blot and immunofluorescence experiments demonstrated that **18** inhibited the NF-κB-activated pathway.

Leptosphin C **21** has been isolated from the solid cultures of an endophytic fungus *Leptosphaeria* sp. XL026 [76]. Its structure was elucidated by extensive spectroscopic methods and single-crystal X-ray diffraction. 

## 4. Fusicoccane

### 4.1. Structural and Biological Diversity

Fusicoccane diterpenoids, characterized by 5/8/5, 5/8/6, 5/9/4, and 5/9/5 fused carbocyclic ring systems, include the fusicoccins, cotylenins, brassicicenes, heterodimers, and homodimers [77,78,79,80]. They were first isolated as glycosides from the phytopathogenic fungus *Fusicoccum amygdali*, in 1964 [81]. Substances exhibiting this structural motif have been isolated from a variety of sources including fungi such as *Talaromyces purpureogenus*, *Alternaria brassicicola* XXC, and *Trichoderma citrinoviride* cf-27 (Figure 3), and rarely from liverworts, algae, ferns, streptomycetes, and higher plants, some of which showed remarkable biological effects relevant for drug discovery, such as antibacterial, antitumor, anti-inflammatory, and antifungal activities [82,83,84,85,86,87,88,89].



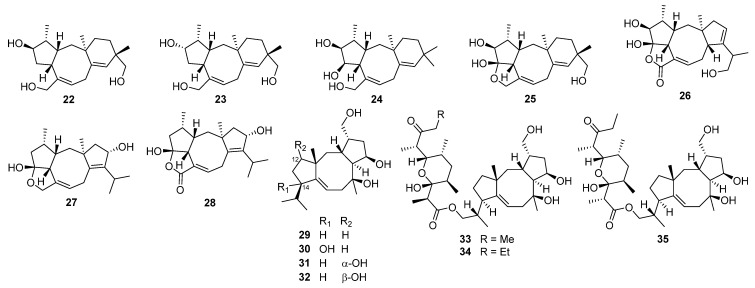



Talaronoids A–D **22**–**25**, four diterpenoids with an unexpected tricyclic 5/8/6 carbon skeleton isolated from *Talaromyces stipitatus*, represent a new class of fusicoccane diterpenoids with a benzo[*a*]cyclopenta[*d*]cyclooctane skeleton [80]. Plausible biosynthetic pathways of talaronoids A–D **22**–**25** have been proposed starting from geranylgeranyl diphosphate with a Wagner–Meerwein rearrangement as the key step (Figure 3). Talaronoids A–D **22**–**25** have shown moderate butyrylcholinesterase (BCHe) inhibitory activity with IC_50_ values of 14.71, 26.47, 31.51, and 11.37 μM, respectively. A new diterpenoid roussoellol C **26** that exhibited moderate antiproliferative activities against human breast adenocarcinoma (MCF-7) cell line with an IC_50_ value of 6.5 μM has been isolated from an extract of laboratory cultures of the marine-derived fungus *Talaromyces purpurogenus* [90]. Roussoellols A **27** and B **28** have been isolated from the plant-inhabiting ascomycetous fungus *Roussoella hysterioides* KT1651 [91]. Compound **28** inhibited the hyphal growth of the phytopathogen *Cochliobolus miyabeanus* at 10 μg/mL. 

Six new fusicoccane-type diterpenoids, 14-hydroxycyclooctatin **30**, 12*α*-hydroxycyclooctatin **31**, 12*β*-hydroxycyclooctatin **32**, fusicomycin A **33**, fusicomycin B **34**, and isofusicomycin A **35**, along with a known compound, cyclooctatin **29** [82,92], have been isolated from the fermentation broth of *Streptomyces violascens* [93]. Compounds **33**–**35** have demonstrated cytotoxicity against five human cancer cell lines (BGC823, H460, HCT116, HeLa, and SMMC7721), with IC_50_ values ranging from 3.5 to 14.1 μM. Cell adhesion, migration, and invasion assays have shown that fusicomycin B **34** inhibited the migration and invasion of human hepatocellular carcinoma SMMC7721 cells in a dose-dependent manner. Through further investigation, it was revealed that **34** inhibited the enzymatic activity of matrix metalloproteinase-2 (MMP-2) and matrix metalloproteinase-9 (MMP-9), in addition to downregulating the expressions of MMP-2 and MMP-9 at both the protein and mRNA levels to influence the migration and invasion of cancer cells.



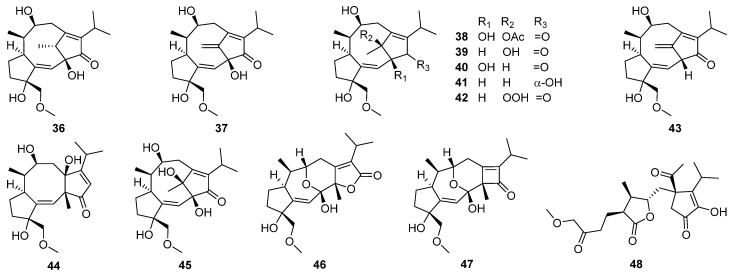



Between 1999 and 2014, eleven new fusicoccane-like diterpenoids were isolated from the phytopathogenic fungus *Alternaria brassicicola* [94,95,96]. With the aid of computational predictions, experimental validation, and biosynthetic logic-based strategies, Zhang and co-workers first rectified the conclusion that all brassicicenes were originally proposed to have a 5/8/5 fused skeleton and, thus, reassigned brassicicenes C–H **36**–**41**, J **42**, and K **43** to have a unique bridgehead double-bond-containing 5/9/5 fused skeleton [97]. Meanwhile, brassicicenes L–N **44**–**46** were three highly modified fusicoccanes also isolated from the fungus *Alternaria brassicicola* [97]. Afterward, alterbrassicene A **47** [78] and alterbrassicicene A **48** [98], two unprecedented fusicoccane-derived diterpenoids featuring a 5/9/4-fused carbocyclic skeleton and a newly transformed monocyclic carbon skeleton (Figure 4), respectively, were obtained from the same fungal strain and found to function on different targets in the NF-κB signaling pathway of anti-inflammatory activity. Later, the biogenetically related intermediates, brassicicenes O **49** and P **50**, were also discovered [78].

Brassicicenes Q–X **51**–**58** have been isolated from the phytopathogenic fungus *Alternaria brassicicola* [99]. Brassicicene S **53** was found to show significant anti-inflammatory activity against the production of NO, TNF-*α*, and IL-1*β* at 10 μM. Further Western blot and immunofluorescence experiments found the mechanism of **53** inhibiting the NF-κB-activated pathway.



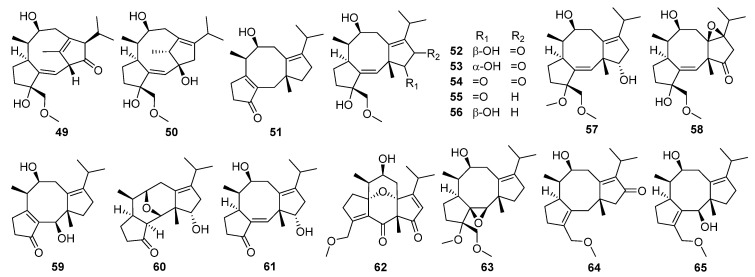



Seven new modified fusicoccane-type diterpenoids **59**–**65**, together with two known congeners, have been obtained from *A. brassicicola* [100]. Alterbrassicicenes B **60** and C **62** represented the first examples of fusicoccane-type diterpenoids featuring two previously undescribed tetracyclic 5/6/6/5 ring systems, while 1*β*,2*β*-epoxybrassicicene I **63** featured a previously undescribed tetracyclic 5/8/5/3 ring system. Alterbrassicicene E **65** showed moderate anti-inflammatory activity against NO production in lipopolysaccharide (LPS)-induced RAW264.7 cell with an IC_50_ value of 24.3 μM. In addition, alterbrassicicene B **60**, 3-ketobrassicicene W **61**, 1*β*,2*β*-epoxybrassicicene I **63**, and alterbrassicicene E **65** exerted weak cytotoxicity against certain human tumor cell lines (OCVAR, MDA-MB-231, HeLa, HT-29, and Hep3B cells) with IC_50_ values ranging from 25.0 to 38.2 μM.



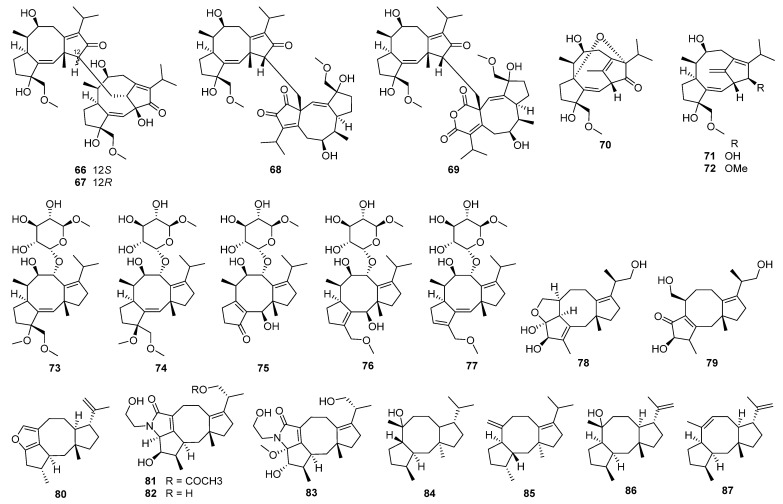



Alterbrassinoids A–D **66**–**69**, the first examples of fusicoccane-derived diterpene dimers furnished by forming an undescribed C-12–C-18′ linkage, have been isolated from modified cultures of *Alternaria brassicicola* [79]. Alterbrassinoids A **66** and B **67** represented unprecedented heterodimers, whereas alterbrassinoids C **68** and D **69** represented unprecedented homodimers, and alterbrassinoid D **69** also featured an undescribed anhydride motif. Alterbrassinoids A–D **66**–**69** showed moderate activities against five cancer cells (including OCVAR, MDAMB-231, HeLa, HT-29, and Hep3B). Afterward, three rearranged fusicoccane diterpenoids bearing a rare bridgehead double-bond-containing tricyclo[9.2.1.0^3,7^]tetradecane (5/9/5 ring system) core skeleton, namely alterbrassicenes B–D **70**–**72**, were obtained from the same fungus *A. brassicicola* [101]. Their structures were assigned via spectroscopic methods, ECD calculations, and single-crystal X-ray diffraction. Compounds **70**–**72** showed moderate cytotoxicity against several human tumor cell lines with IC_50_ values ranging from 15.87 to 36.85 μM.

Five new diterpenoid glycosides, dongtingnoids A–E **73**–**77**, two new diterpenoid aglycones, dongtingnoids F **78** and G **79**, and two known analogues, cotylenins E and J, belonging to the fusicoccane family, have been isolated from the fungus *Penicillium* sp. DT10 [102]. Dongtingnoids A **73**, D **76**, and E **77** showed comparable seed-germination-promoting activities to the growth regulator cotylenin E [103,104]. Such diterpene glucosides have been used for the production of an intermediate compound suitable for semi-synthesis by a mutant constructed by disruption of a specific gene by homologous recombination [105,106].

Trichocitrin **80**, representing the first *Trichoderma*-derived and furan-bearing fusicoccane diterpene, has been isolated from the culture of marine brown alga-endophytic *Trichoderma citrinoviride* [107]. A new class of fusicoccane-type diterpenoid alkaloids with an unusual 5/5/8/5 tetracyclic system, i.e., pericolactines A–C **81**–**83**, have been isolated from *Periconia* sp. [108].

### 4.2. Biosynthesis of Fusicoccane Diterpenes

A unique chimeric enzyme PaFS, possessing both a geranylgeranyl diphosphate (GGDP) synthase domain and a diterpene cyclase domain, has been identified from *Phomopsis amygdali* [109]. A biosynthetic gene cluster of brassicicene C **36**, a fusicoccadiene synthase (AbFS) containing 11 genes (*orf1* to *orf11*, Figure 5A), has been identified in *Alternaria brassicicola* ATCC 96836 from genome database search [110,111]. In vivo and in vitro studies have clearly revealed the function of *Orf8* and *Orf6* as a fusicoccadiene synthase similar to PaFS and methyltransferase, respectively. In this gene cluster, five genes (*orf1*, *orf2*, *orf5*, *orf7*, and *orf11*) encoded cytochrome P450s. *Orf9* was a key dioxygenase to determine the aglycon structures of fusicoccin and brassicicene [112].

Other fusicoccane-type diterpene synthases have been identified from bacteria or fungus, such as CotB2 from bacteria responsible for the biosynthesis of cyclooctat-9-en-7-ol **84** [113], and SdnA from fungus responsible for the biosynthesis of cycloaraneosene **85** [114]. The same 5/8/5 tricyclic skeleton occurred in the sesterterpene ophiobolin F for which the terpene synthase AcOS has been reported from *Aspergillus clavatus* [115]. Oikawa and co-workers applied the *Aspergillus oryzae* heterologous expression system to functionally characterize cryptic bifunctional terpene synthase genes found in fungal genomes and identified the sesterfisherol (contains a characteristic 5/6/8/5 tetracyclic system) synthase gene (*NfSS*) from *Neosartorya fischeri* [116].

A unique P450 enzyme *bscF* has been identified in the phytopathogen *Pseudocercospora fijiensis* that generated two structurally different products from the single substrate. In addition to the heterologous expression of the eight genes, *bscA*-*bscH* elucidated the biosynthetic pathway for brassicicenes (Figure 5B) [117]. 

A new fusicoccane-type diterpene synthase MgMS has been identified from the fungus *Myrothecium graminearum* by the genome mining method, which catalyzed the formation of the new diterpene alcohol myrothec-15(17)-en-7-ol **86** with all the seven stereocenters being introduced in the cyclization steps and conserved in the structure of the product. Based on this, its novel cyclization mode was unambiguously assigned (Figure 6) [118].

## 5. Guanacastane

The discovery of 5/7/6 ring-fused guanacastane diterpenoids has been limited to several fungal species in the different genera *Cercospora*, *Cortinarius*, *Coprinellus*, *Coprinus*, *Psathyrella*, and *Verticillium* (Figure 4). *Coprinellus heptemerus* and *C. radians* M65 belong to the same genus, and first, they gather into one branch. *Psathyrella candolleana* and *Cercospora* sp. gather into one branch although they come from different genera. They are all able to produce guanacastane diterpenoids, indicating that highly homologous BGCs may also exist in fungi of different genera. The first member guanacastepene A **88**, a new diterpene antibiotic against methicillin-resistant *Staphylococcus aureus* (MRSA) and vancomycin-resistant *Enterococcus faecalis* (VREF), has been isolated from an unidentified endophytic fungus [119]. Meanwhile, fourteen new analogues guanacastepenes B–O **89**–**102** have been isolated from the same resource [120]. The novel skeleton has attracted great interests for organic synthesis [121,122,123,124,125,126,127,128,129,130,131,132,133,134,135,136]. The biological activities of guanacastanes have mainly been reported to possess cytotoxicity and antimicrobial effects.



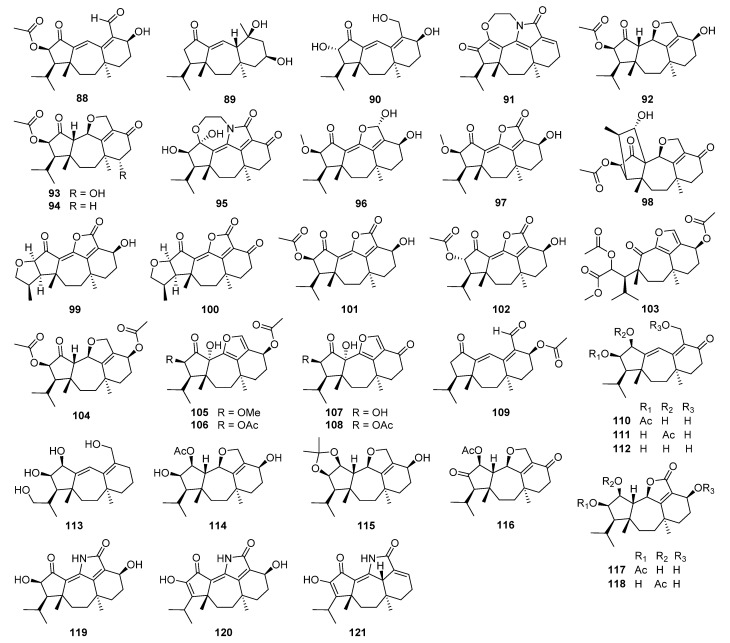



Heptemerones A–G **103**–**109** have been isolated from cultures of *Coprinus heptemerus* [137,138]. Radianspenes A–M **110**–**122** have been obtained from *Coprinus radians* [139]. Among the biological activities of these isolates, the inhibition of fungal germination was the most potent, and depended highly on the composition of the assay medium [137]. Radianspene C **112** showed inhibitory activity against human breast carcinoma (MDA-MB-435) cell with an IC_50_ value of 0.91 μM [139]. Investigation of secondary metabolites from the fungal *Coprinus plicatilis* led to the discovery of several new guanacastane-type diterpenoids, named plicatilisins A–D **123**–**126** [140] and E–H **127**–**130** [141]. In vitro cytotoxic activities against the human cancer cell lines (HepG2, HeLa, MDA-MB-231, BGC-823, HCT 116, and U2OS) showed that plicatilisin A **123** exhibited significant cytotoxicity with IC_50_ values ranging from 1.2 to 6.0 μM [140].

Guanacastepenes P–T **131**–**135** have been isolated from cultures of the fungus *Psathyrella candolleana* [142]. Guanacastepene R **133** exhibited inhibitory activity against both human and mouse isozymes of 11*β*-hydroxysteroid dehydrogenase (11*β*-HSD1) with IC_50_ values of 6.2 and 13.9 μM, respectively. Cercosporenes A–F **136**–**141**, including two homodimers **140** and **141**, have been isolated from the fungus *Cercospora* sp. [143]. Cercosporene F **141** was cytotoxic to five human tumor cell lines (HeLa, A549, MCF-7, HCT116, and T24) with IC_50_ values of 8.16–46.1 μM, and induced autophagy in HCT116 cell.



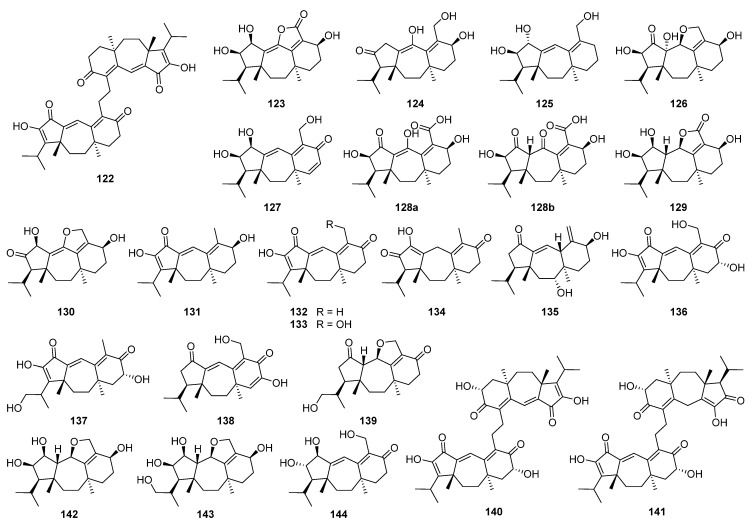



Eleven new guanacastane-type diterpenoids dahlianes A–K **142**–**152** have been obtained from the fungus *Verticillium dahlia* that was isolated from the gut of insect *Coridius chinensis* [144,145]. In the cytotoxicity evaluation against human tumor cell lines, dahlianes B **143** and C **144** exhibited significant cytotoxicity against human breast cancer cell MCF-7 with IC_50_ values of 3.35 and 4.72 μM, respectively [144]. In addition, the isolates were evaluated for their cytotoxicity toward drug-sensitive and DOX resistant MCF-7 cells by MTT assay. As a result, dahliane G **148** showed an 80-fold potentiation effect on the sensitization of doxorubicin at the concentration of 15 μM when screening the reversal activity on doxorubicin-resistant human breast cancer cell (MCF-7/DOX) [145].

Pyromyxones A–D **153**–**156** have been isolated from fruiting bodies of *Cortinarius pyromyxa*, which possessed an undescribed *nor*-guanacastane skeleton of a 5/7/6 tricyclic system [146]. Pyromyxones A **153**, B **154**, and D **156** exhibited weak activity against Gram-positive *Bacillus subtilis* and Gram-negative *Aliivibrio fischeri,* as well as the phytopathogenic fungi *Botrytis cinerea*, *Septoria tritici,* and *Phytophthora infestans* [146].



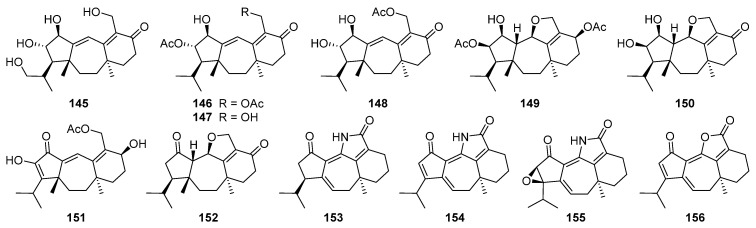



## 6. Harziene

Harziene is a small group of diterpenoids that have a unique 4/7/5/6 tetracyclic scaffold. They have mainly been obtained from different *Trichoderma* species and rarely from liverworts [147]. Harziandione **157** was the first harziene diterpenoid isolated from the liquid culture of *T. harzianum,* in 1992 [148]. Harzianone **158**, a new harziene diterpene, has been isolated from an alga-endophytic isolate of *T. longibrachiatum* [149]. The structure with absolute configuration of **158** was unambiguously identified by NMR and mass spectrometric methods as well as quantum chemical calculations. In addition, the absolute configuration of harziandione **157** was supported by optical rotation calculation, and the structure of isoharziandione isolated from culture filtrate of a strain of *Trichoderma viride* [150] was revised to harziandione **157** on the basis of ^13^C NMR data comparison and calculation.



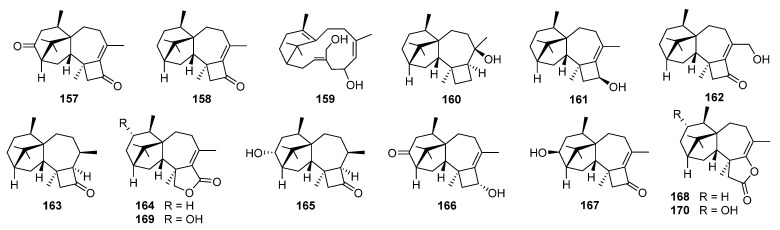



The terpene cyclization mechanism of harzianone **158** has been studied by feeding experiments using selectively ^13^C- and ^2^H-labeled synthetic mevalonolactone isotopologues, followed by the analysis of the incorporation patterns of ^13^C NMR spectroscopy and GC/MS, and the structure of harzianone **158** was further supported from a ^13^C-^13^C COSY experiment of the in vivo generated fully ^13^C-labeled diterpenoid (Figure 7) [151].

Four new harziene-related compounds **159**–**162** have been isolated from an endophytic fungus *Trichoderma atroviridae* UB-LMA [152]. Among them, **159** is a potential derivative of geranylgeranyl diphosphate and may represent the biosynthetic precursor of this scarce family of compounds (Figure 7). Recently, the first total synthesis of nominal harziene diterpenoid **160** has been achieved; stereochemical analysis and subsequent synthesis of the epimeric tertiary alcohol led to the reassignment of configuration for compound **160** as shown for harzianol I **180** [153].

(9*R*,10*R*)-Dihydro-harzianone **163** and harzianelactone **164** have been isolated from the endophytic fungus *Trichoderma* sp. Xy24 [154]. Compound **163** was the reductive product of harzianone **158** while **164** possessed a 6/5/7/5-fused ring core containing a lactone. The latter was the Baeyer–Villiger monooxygenase catalyzed oxidation product of harzianone **158**. Compound **163** exhibited cytotoxicity against HeLa and MCF-7 cell lines with IC_50_ values of 30.1 and 30.7 μM, respectively.

3*R*-Hydroxy-9*R*,10*R*-dihydroharzianone **165** has been isolated from an endophytic fungus *Trichoderma harzianum* X-5 [155]. 11-Hydroxy-9-harzien-3-one **166**, isolated from *T. asperellum* cf44-2, showed inhibitory activity against pathogenic bacteria *Vibrio parahaemolyticus* with a 6.2 mm zone [156]. 3*S*-Hydroxyharzianone **167**, isolated from *T. asperellum* A-YMD-9-2, could highly inhibit four marine phytoplankton species (*Chattonella marina*, *Heterosigma akashiwo*, *Karlodinium veneficum*, and *Prorocentrum donghaiense*) with the IC_50_ values ranging from 3.1 to 7.7 μg/mL [157]. Deoxytrichodermaerin **168**, a harziene lactone possessing potent inhibition against the four phytoplankton species (*C. marina*, *H. akashiwo*, *K. veneficum*, and *P. donghaiense*), has been obtained from an endophyte *Trichoderma longibrachiatum* A-WH-20-2 [158].

Two new harziene diterpene lactones, i.e., harzianelactones A **169** and B **170**, and five new ones, i.e., harzianones A–D **171**–**174** and harziane **175**, have been isolated from the soft coral-derived fungus *Trichoderma harzianum* XS-20090075 [159]. These compounds exhibited potent phytotoxicity against seedling growth of amaranth and lettuce. Harzianone E **176**, which exhibited weak antibacterial activity against *Photobacterium angustum*, has been obtained from the culture of coral-derived fungus *T. harzianum* treated with 10 µM sodium butyrate [160]. Harzianols F–J **177**–**181** and three known derivatives have been obtained from the liquid fermentation of an endophytic fungus *T. atroviride* B7 [161]. Among them, compound **180** exhibited significant antibacterial effect against *Staphylococcus aureus*, *Bacillus subtilis*, and *Micrococcus luteus* with EC_50_ values of 7.7, 7.7, and 9.9 μg/mL, respectively. Meanwhile, cytotoxic activity of **180** against three cancer cell lines was also observed [161].

Furanharzianones A **182** and B **183** are two new harziene-type diterpenoids with an unusual 4/7/5/6/5 ring system, while harzianols A–E **184**–**188** and harziane acid **189** are six new oxidized derivatives of harzianone [162,163]. These compounds have all been obtained from microbial transformation by the bacterial strain *Bacillus* sp. IMM-006.



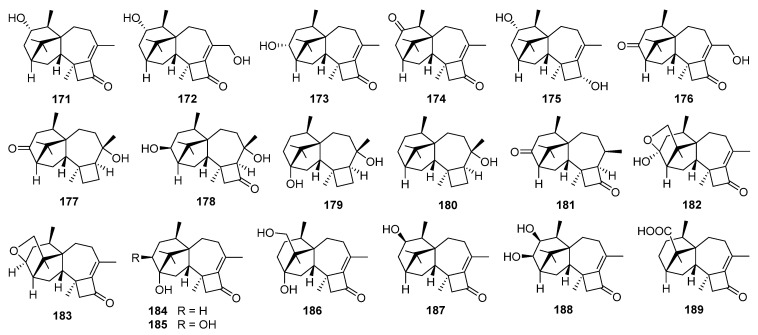



## 7. Phomopsene

Structure determination of the novel diterpene hydrocarbon phomopsene **190** has been provided by enzymatic synthesis with the recombinant terpene synthase PaPS from *Phomopsis amygdali,* and screening of fungal broth extracts regarding characteristic NMR signals of phomopsene **190** resulted in the isolation of a new diterpene, methyl phomopsenonate **191** (Figure 8) [164].

The cyclization mechanism of tetracyclic diterpene phomopsene **190** with phomopsene synthase (PaPS) has been examined through systematically deuterium-labeled geranylgeranyl diphosphate (GGPP), starting from site-specific deuterium-labeled isopentenyl diphosphates (IPPs) using IPP isomerase and three prenyltransferases (Figure 9) [165].

Otherwise, other phomopsene synthases have been identified from actinomycetes such as *Allokutzneria albata* (PmS), *Nocardia testacea* (NtPS), and *Nocardia rhamnosiphila* (NrPS) [166,167]. All enzymes were subjected to in-depth mechanistic studies involving isotopic labeling experiments, metal-cofactor variation, and site-directed mutagenesis.



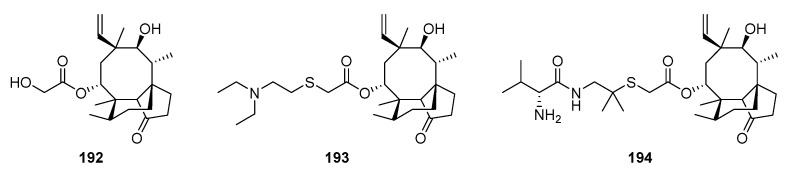



## 8. Pleuromutilin

Pleuromutilin **192** is a diterpene with a tricyclic skeleton possessing antimicrobial properties. It was first discovered from two basidiomycete fungal species including *Pleurotus mutilis* (synonymous to *Clitopilus scyphoides* f. *mutilus*) and *Pleurotus passeckerianus* (synonymous to *Clitopilus passeckerianus*) [168], and then produced by a number of other related species [169]. Its chemical structure and cyclisation mechanism has been elucidated by independent works [170,171,172], while total synthesis has been achieved by [173,174]. The semi-synthetic pleuromutilin analogues tiamulin **193** and valnemulin **194** have been used for over three decades to treat economically important infections in swine and poultry without showing any significant development of resistance in their target bacteria [21,22,23,24,25]. In recent years, extensive research including structure–activity relationship studies have been conducted to generate new orally available pleuromutilin derivatives having been used systemically in human medicine to treat acute bacterial skin and skin structure infections, as well as multidrug-resistant tuberculosis [175,176,177,178].

The gene cluster for the antibiotic pleuromutilin **192** has been isolated in *Clitopilus passeckerianus* [179]. Total de novo biosynthesis of pleuromutilin **192** was achieved through the expression of the entire gene cluster in the secondary host *Aspergillus oryzae*, proving that the seven genes isolated were sufficient for biosynthesis of the diterpene antibiotic. Heterologous expression of genes from the pleuromutilin gene cluster in *A. oryzae* revealed the biosynthesis of the antibiotic pleuromutilin **192** (Figure 10) and generated bioactive semi-synthetic derivatives [180].



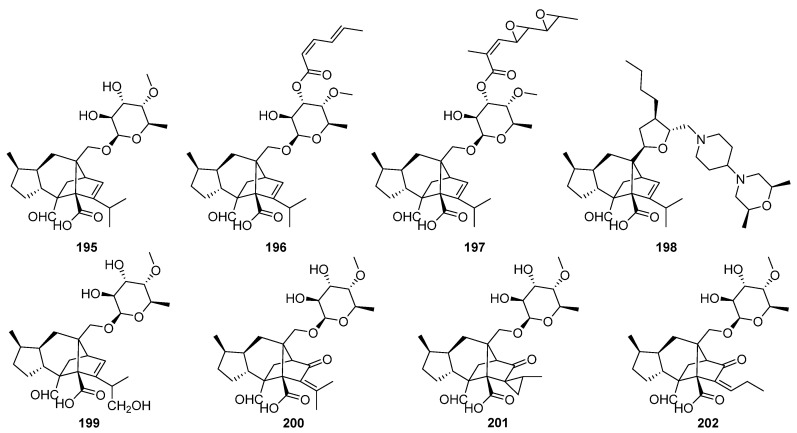



## 9. Sordaricin

Sordarin **195**, an antifungal antibiotic possessing a unique 5/6/5/5-fused ring system, was discovered in 1971 as a metabolite of *Sordaria araneosa* [181]. A number of related semisynthetic sordarin derivatives have also been reported and some have been developed as antifungal agents such as zofimarin **196**, hypoxysordarin (FR231956) **197**, and FR290581 **198** [182,183,184,185,186]. Sordarin **195** and related compounds have been shown to inhibit protein synthesis by a mechanism involving selective binding to the elongation factor 2 (EF-2) and ribosome complex in fungi [187,188,189]. 

Sordarins C–F **199**–**202**, possessing a unique 5/6/5/5 or 5/6/5/5/3 ring system varied at the C-11 position and the branch attached to C-12 of the sordaricin-type diterpene skeleton, have been isolated from the fungus *Xylotumulus gibbisporus* [190]. Genome mining of the sordarin biosynthetic gene cluster from *Sordaria araneosa* has been carried out, and the results suggest that the identified *sdn* gene cluster is responsible for the biosynthesis of sordarin **195** and hypoxysordarin **197** (Figure 11) [114].

## 10. Tetraquinane

Several antibiotic crinipellin-related diterpenoids containing a 5/5/5/5 tetraquinane skeleton have been obtained from the basidiomycetous fungus *Crinipellis stipitaria* [191,192]. Up to now, the total synthesis of (±)-crinipellin B **203** and (–)-crinipellin A **204** have been reported [186,193,194,195].

Four novel diterpenoids, namely (4*β*)-4,4-*O*-dihydrocrinipellin A **205**, (4*β*,8*α*)-4,4-*O*,8,8-*O*-tetrahydrocrinipellin B **206**, crinipellins C **207** and D **208**, along with three known diterpenoids have been isolated from the fungus *Crinipellis* sp. 113 [196]. Antitumor assays demonstrated that the compounds possess moderate activities against HeLa cell. 

Four new tetraquinane diterpenoids crinipellins E–H **209**–**212** have been isolated from fermentations of a *Crinipellis* species [197]. Crinipellins E–G **209**–**211** inhibited the LPS/IFN-γ induced CXCL10 promoter activity in transiently transfected human MonoMac6 cell in a dose-dependent manner with IC_50_ values of 15, 1.5, and 3.15 μM, respectively. Moreover, crinipellins E–G **209**–**211** reduced mRNA level and synthesis of proinflammatory mediators such as cytokines and chemokines in LPS/IFN-γ stimulated MonoMac6 cell.

A new crinipellin derivative crinipellin I **213** together with the known crinipellin A **204** have been obtained from the fungus *Crinipellis rhizomaticola* [198]. Crinipellin A **204** exhibited a wide range of antifungal activity in vitro against *Colletotrichum coccodes*, *Magnaporthe oryzae*, *Botrytis cinerea*, and *Phytophthora infestans* (MICs of 1, 8, 31, and 31 µg/mL, respectively).



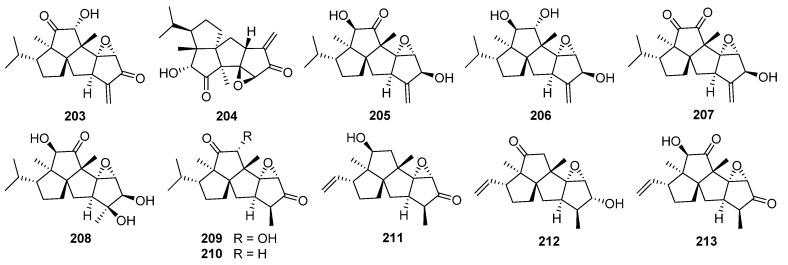



## 11. Others

### 11.1. Spirograterpene

A novel spiro-tetracyclic diterpene featuring a 5/5/5/5 spirocyclic carbon skeleton, i.e., spirograterpene A **214**, has been isolated from the deep-sea-derived fungus *Penicillium granulatum* [199]. Spiroviolene **215**, bearing the same carbon skeleton to that of **214**, has been obtained from a bacterial terpene synthase [200]. Spirograterpene A **214** showed an antiallergic effect on immunoglobulin E (IgE)-mediated rat mast RBL-2H3 cell with 18% inhibition as comparedwitho 35% inhibition for the positive control (loratadine) at the same concentration of 20 μg/mL [199].



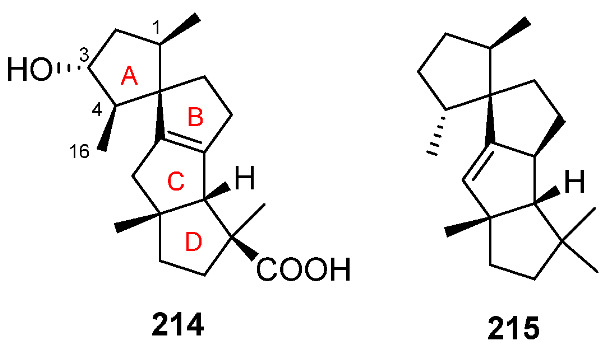



### 11.2. Psathyrin

Two skeletally novel tetracyclic diterpenoids that possess a novel 5/5/4/6 tetracyclic system, psathyrins A **216** and B **217**, have been characterized from cultures of the basidiomycete *Psathyrella candolleana*. They displayed weak antibacterial activities against *Staphylococcus aureus* and *Salmonella enterica*. The biosynthetic pathway of **216** and **217** was proposed to start from GGPP and the final products were obtained through a series of reactions (Figure 12) [201].

### 11.3. Coicenal

Coicenals A–D **218**–**221**, possessing a previously undescribed 6/6 fused carbon skeleton, have been isolated from the solid culture of the plant pathogenic fungus *Bipolaris coicis* [202]. Coicenals A **218** and B **219** could be transformed into **221** and compound **222** by treatment with acetyl chloride, respectively. Coicenals A–D **218**–**221** showed moderate inhibitory activity against NO release with IC_50_ values of 16.34, 23.55, 10.82, and 54.20 µM, respectively.



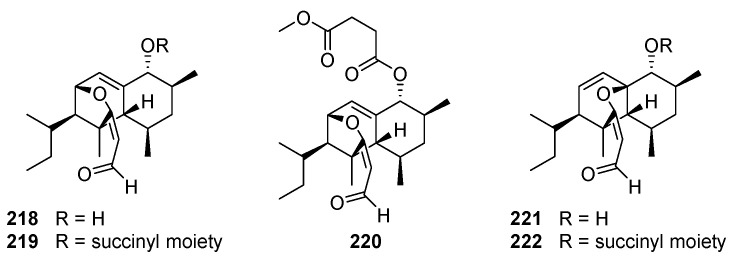



### 11.4. Eryngiolide

Eryngiolide A **223** has been isolated from the solid culture of the edible mushroom *Pleurotus eryngii* [203]. It is the first member of diterpenoids with a novel skeleton deriving from a cyclododecane core fused with two *γ*-lactone units [203]. It has exhibited moderate cytotoxicity against two human cancer cell lines (Hela and HepG2) in vitro. Biogenetically, eryngiolide A **223** could be the first diterpene not synthesized from GGPP unit, which indicated a completely new route for diterpene biosynthesis in nature (Figure 13).

### 11.5. Trichodermanin

Trichodermanin A **224**, a structurally unique diterpenoid with skeletal carbons arranged compactly in a 6/5/6/6 ring system, has been isolated from cultures of *Trichoderma atroviride* [204]. Its absolute configuration was elucidated by single crystal X-ray diffraction. Wickerols A **225** and B **226** were two novel diterpenoids produced by *Trichoderma atroviride* and the absolute configuration of **226** was confirmed by X-ray crystallographic analysis [205,206]. Wickerol A **225** showed potent antiviral activity against the A/H1N1 flu virus (A/PR/8/34 and A/WSN/33 strains) with an IC_50_ value of 0.07 μg/mL, but not active against the A/H3N2 virus. Wickerol B **226** also showed anti-influenza virus activity against A/PR/8/34 virus with an IC_50_ value of 5.0 μg/mL [206].



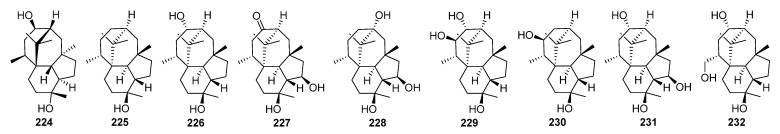



The new skeleton of wickerols A **225** and B **226** was revealed by the feeding experiments of [1-^13^C]-, [2-^13^C]-, and [1,2-^13^C_2_]-acetates, respectively [206]. The cyclization mechanism of wickerol B **226** was predicted, as shown in Figure 14. First, pyrophosphate was ejected from the terminus of the boat-like transition state of GGPP, forming a verticillen-12-yl cation intermediate, the same as the first step of phomactatriene and taxadiene biosynthesis [207]. 1,2-Rearrangements of *β*-methyl and *α*-hydride occurred at the six-membered ring part, then, the ring inversion and cyclization progressed to form the 6/5/9 ring intermediate. A rearrangement proceeded to expand the ring from five to six membered, and the last step resulted in the formation of the 6/5/6/6 ring skeleton. The C-8 position of wickerol A **225** was oxidized by a cytochrome P450 to give wickerol B **226**.

Trichodermanins C–H **227**–**232** are new diterpenes with a 6/5/6/6 tetracyclic system that have been isolated from the marine sponge-derived fungus *Trichoderma harzianum* [208,209]. Trichodermanin C **227** potently inhibited the growth of murine P388 leukemia, human HL-60 leukemia, and murine L1210 leukemia cell lines with IC_50_ values of 7.9, 6.8, and 7.6 μM, respectively [208].

## 12. Conclusions and Future Prospects

Diterpenoids show huge potential for drug discovery and development due to their extensive biological functions and structural diversity. Fungal diterpenoids are a diverse family of hybrid natural products with potent bioactivities and intriguing structural architectures. A large number of fungal diterpenoids have exhibited significant anti-inflammatory, cytotoxic, anti-MRSA, antimicrobial, antiviral, antihypertensive, and platelet aggregation-inhibitory activities. Consequently, these bioactive diterpenoids are always hot trending topics for the synthesis community [173,174,186]. Nevertheless, the structural complexity and limited availability of natural products remain obstacles to synthesizing a large collection of natural products and their structural analogues in sufficient amounts. Thus, a synthetic biology method based on the combination of heterologous biosynthesis and genome mining is a promising approach to translate enormous amounts of biosynthetic gene information to richly diverse natural products. Interestingly, while fungi have evolved their systems to create terpenoid diversity, they have also biosynthesized some of the same classes of terpenoids found in plants, bacteria, and other organisms. These relationships provide accessible and renewable prokaryotic systems for eukaryotic natural product biosynthesis and enzymology. In conclusion, we hope it is evident from this review that most of the fungal diterpenoids are biologically active with a few key scaffolds paving a path towards potential drug discovery and development.

## Data Availability

Not applicable.

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
