# Peer review of "Diterpenes Specially Produced by Fungi: Structures, Biological Activities, and Biosynthesis (2010–2020)"

_jof, 2022, doi:10.3390/jof8030244_

Round 1

Reviewer 1 Report

The authors well summarized diterpene data, particularly their structures classified by their skeletons, in this review. And the three charts in Introduction appeared very helpful for the readers. I have several comments to be considered by the authors.

Major points:

  1. The 4 figures showed phylogenetic analysis of the ITS sequences from each diterpene-producing fungi selected. What did the authors explain from the data? It will be more worthy if the figure shows the diterpenes (the compound number in the  text)produced by the fungus. Data from selected producing-fungi should be most (all) producing-fungi  in the text.
  2. Recently, data for BGC of secondary metabolites including fungal diterpenes are accumulating. The authors showed such data in Scheme 5A. Can the authors collect the sequence data of BGC involving diterpene synthases and/or cyclases appeared in the text  to make phylogenetic tree?

Author Response

  1. The 4 figures showed phylogenetic analysis of the ITS sequences from each diterpene-producing fungi selected. What did the authors explain from the data? It will be more worthy if the figure shows the diterpenes (the compound number in the text) produced by the fungus. Data from selected producing-fungi should be most (all) producing-fungi in the text.

Re. The phylogenetic analysis results showed that source genera of fungal diterpenoid are grouped based on their regiospecificity. On the other hand, since this paper only focuses on new compounds, there is no statistics on the total amount of specific diterpenoids (including known compounds) produced by a certain fungus.

  1. Recently, data for BGC of secondary metabolites including fungal diterpenes are accumulating. The authors showed such data in Scheme 5A. Can the authors collect the sequence data of BGC involving diterpene synthases and/or cyclases appeared in the text to make phylogenetic tree?

Re. This is a good suggestion. However, we currently do not have enough energy to deal with this matter, and at the same time we do not have the relevant professional level for fear of making some mistakes. This review focuses on the chemistry, biological functions and synthetic biology of fungal natural diterpenoids, meeting the requirements of the special issue. Therefore, we regret not being able to make this revision.

Reviewer 2 Report

The review article is very good, it is an extensive revision on the subject, mainly from the last 30 years, and the authors mention the search criteria of the subject matter. The chemical structures are fine. However, I suggested that the authors present them in Figures format, since these are found without context throughout the text.

The methods and results of others are adequately presented by the authors.

The English is clear and the work, as a whole, is clearly written.

My suggestion was to accept it after making minor changes, among which I suggest rearranging the chemical structures, to make this long review easier to read.

Author Response

My suggestion was to accept it after making minor changes, among which I suggest rearranging the chemical structures, to make this long review easier to read.

Re. We have readjusted the layout of the chemical structures which have been highlighted in the revised manuscript.